# Compounding Tailored Veterinary Chewable Tablets Close to the Point-of-Care by Means of 3D Printing

**DOI:** 10.3390/pharmaceutics14071339

**Published:** 2022-06-24

**Authors:** Erica Sjöholm, Rathna Mathiyalagan, Xiaoju Wang, Niklas Sandler

**Affiliations:** 1Pharmaceutical Sciences Laboratory, Faculty of Science and Engineering, Åbo Akademi University, Tykistökatu 6A, 20520 Turku, Finland; rathna.mathiyalagan@abo.fi (R.M.); xiaoju.wang@abo.fi (X.W.); niklas.o.sandler@gmail.com (N.S.); 2Nanoform Finland Oyj, Viikinkaari 4, 00790 Helsinki, Finland; 3CurifyLabs Oy, Kaikukatu 4A, 00530 Helsinki, Finland

**Keywords:** semi-solid extrusion 3D printing, additive manufacturing, compounding, drug delivery, tailored dosage forms, veterinary medicine, theophylline, chewable tablet

## Abstract

Certain patient populations receive insufficient medicinal treatment due to a lack of commercially available products. The number of approved veterinary products is limited, making animals a patient population with suboptimal medicinal treatments available. To answer to this unmet need, compounding and off-label use of human-marketed products are practiced. Both of which have a significant risk of preparation errors. Hence, there is a dire demand to find and implement a more automated approach to the accurate, precise, and rapid production of veterinary dosage forms close to the point-of-care. This study aimed to assess the use of semi-solid extrusion-based 3D printing for the preparation of tailored doses of theophylline in the form of a chewable dosage form suitable for veterinary use. This study proved that semi-solid extrusion-based 3D printing could successfully be utilized to manufacture pet-friendly, chewable theophylline-loaded tablets. The prepared dosage forms showed a high correlation (R^2^ = 0.9973) between the designed size and obtained drug amount and met the USP and Ph. Eur. content uniformity criteria. Furthermore, the stability study showed the dosage form being stable and able to be used for up to three months after printing.

## 1. Introduction

There are a limited number of commercially available veterinary medicinal products. Hence, compounding has been a common practice to fill therapeutic gaps and fulfill the needs that currently exist in the veterinary landscape. The compounding practice involves altering an approved medicinal product to achieve an acceptable dosage form or increasing the adherence by adding flavor. It also consists of the preparation of the desired product from pure substances [1]. Theophylline (TPH) is a widely used bronchodilator for the treatment of asthma and chronic obstructive pulmonary disease in humans [2]. It is also currently used off-label for the treatment of asthma in cats and dogs due to the lack of approved veterinary TPH products [2,3], but the marketed doses are often too high to suit the therapeutic needs of animals. Furthermore, animals are not small humans, and therefore human dosage forms cannot always be scaled down according to the weight of the animal [4], nor does the dosage form allow for precise down-scaling. There are also other aspects to consider when developing veterinary formulations. Some excipients used in human medicinal products are toxic to animals, and the dosage form or route of administration may not be suitable for animals [1]. In addition, the anatomical difference between humans and animals must be taken into consideration; for instance, capsules and tablets can get stuck in a horizontal esophagus, and animals are prone to spit out solid dosage forms. Hence, oral solutions and chewable tablets (ChewTs) are common on the veterinary drug market.

In contrast to approved drugs that are prepared and tested in accordance with good manufacturing practices, compounded medicines are exempt from this, consequently leading to inconsistent quality. Preparation errors are relatively common; at their worst, they can lead to a lack of effect or lethal overdoses. In addition, compounded medicines are made manually in pharmacies, making it a highly labor-intensive practice. Furthermore, contaminations, physical and chemical instability of the finished product, and a lack of bioavailability in the target animal are also points of concern [1]. Hence, there is a need for a more automated approach to reduce the burden of compounded medicines while rendering higher quality products on-demand. 

The 3D printing technologies have been investigated as a means of on-demand production of tailored medicinal products to fit the need of specific patient populations. It is considered that 3D printing has a great potential to create new therapies and improve the adherence, safety, and efficacy of existing therapies [5]. Several studies have been conducted where TPH-containing dosage forms have been produced by coupling hot-melt extrusion and fused deposition modeling 3D printing. TPH-containing immediate-release tablets [6,7], gastro-retentive floating tablets [8,9], immediate and extended-release tablets [10], sustained-release tablets [11,12], and delayed-release tablets [13] have been explored. Other extrusion-based 3D printing techniques have also been investigated. Anderspuk et al. developed a TPH tablet using micro-extrusion-based 3D printing and investigated the crosslinking effects on the physical solid-state and its dissolution properties [14]. Cheng et al. prepared TPH-containing extended-release tablets using semi-solid extrusion (SSE)-based 3D printing, and they investigated the effect of the ink’s rheological properties on the printability [11]. Dores et al. manufactured immediate release TPH tablets using a novel temperature- and solvent-facilitated extrusion-based direct ink writing 3D printing technique [15]. Furthermore, Kuzminska et al. prepared TPH-containing oral tablets using the solvent-free direct printing ink writing method, eliminating the post-printing drying step associated with conventional SSE 3D printing [16].

As in all 3D printing techniques, the desired structure is built up in a layer-by-layer approach. However, the starting material used in SSE 3D printing differs from other extrusion-based printing techniques, as semi-solid or semi-molten materials are used instead of solid filaments or powders, which facilitates the employment of low printing temperatures, making it a more suitable technique for the production of medicinal products and biomedical applications. Furthermore, the material is loaded into disposable syringes providing benefits in meeting critical quality requirements for pharmaceutical use [17]. Lately, SSE 3D printing has gained interest as a manufacturing technique for the production of drug-loaded dosage forms. In addition to the TPH-containing dosage forms described above, the technique has been explored for the production of a variety of drug delivery systems, such as tablets with high drug-loading [18], gastric floating core-shell structured tablets with sustained-release [19], ophthalmologic patches with controlled drug release [20], cereal-based tablets for pediatric use [21], solid lipid tablets [22], chewable tablets [23,24,25,26], personalized orodispersible films [27,28,29], and suppositories [30].

Utilizing 3D printing to produce veterinary TPH products has, to the best of the authors’ knowledge, not previously been explored. However, SSE 3D printing has previously been investigated in the preparation of veterinary medicinal products containing prednisolone [28] and gabapentin (GBP) [26]. The current study aimed to examine if the same printing ink base used in the GBP study could be utilized with a different drug, namely TPH. Our intention was not only to simplify the SSE 3D printing by providing one base suitable for preparation of several veterinary drug products on-demand, but also to investigate the drug’s influence on the dosage form. Furthermore, the stability of the prepared ChewTs was investigated over a three-month period to explore the shelf life of the prepared product. The study’s overarching goal was to explore the use of SSE 3D printing as an automated approach to the on-demand production of tailored ChewTs suitable for veterinary use close to the point-of-care.

## 2. Materials

Theophylline (TPH) was used as the active pharmaceutical ingredient in the current study. It was obtained from Fluka Biochemika, Sigma Aldrich (Steinheim, Germany), and used as received. Hydroxypropyl methylcellulose (HPMC, Methocel K3 Premium LV) was kindly donated by Dow Chemical Company (Bomlitz, Germany) and used as a film-forming agent. The super disintegrant crospovidone (Kollidon CL), kindly provided by BASF (Ludwigshafen, Germany), was added to the formulation to decrease the disintegration time. In the formulation, mannitol (Ph. Eur., Merck, Darmstadt, Germany) was used as a filling agent, 85% aqueous solution of glycerol (Fagron, Barsbüttel, Germany) was used as a plasticizer, and pure liver powder (LP, CC Moore & Co., Stalbridge, UK) was added as a taste-enhancer to make the chewable tablet more palatable. Purified water (MQ, Milli-Q^®^, Merck Millipore, Molsheim, France) was used as solvent in the formulation and in all the analyses except for the measurement of the salivary pH, which was performed in sterile water (sterilized water by Fresenius Kabi Norway AS, Halden, Norway).

## 3. Methods

### 3.1. Ink Preparation

The ink in the current study was prepared in the same manner as the formulation made by Sjöholm et al. [26] in order to study the ability to utilize the same ink base in the preparation of drug-loaded inks containing different drugs. Hence, the only difference in the preparation was the addition of the active pharmaceutical ingredient (API). Gabapentin (GBP) was used in the previous study, while theophylline (TPH) was investigated in the current study. For the preparation of the printing ink, firstly, a 15% (*w/v*) aqueous hydroxypropyl methylcellulose (HPMC) gel was prepared and kept in the fridge overnight. The following day the printing ink was prepared by mortar and pestle. The dry powders, 20% TPH, 27% mannitol, 6% crospovidone, and 3% liver powder (LP), were first mixed upon 8% glycerol, 35% pre-made HPMC gel, and 1% purified water (MQ) was added. After thoroughly mixing, a brown paste was obtained. The prepared printing ink was left to stand in 10 mL optimum clear barrel syringes with optimum clear pistons (both by Nordson EFD LLC, East Providence, RI, USA) for two hours prior to printing to remove any bubbles as well as let the ink settle as found in rheology results obtained in the previous study [26]. A drug-free placebo ink was prepared in the same manner as the drug-loaded ink without the addition of the API and was used as a reference in the performed analysis.

### 3.2. Printing Ink Characterization

Rheology measurement and solid-state characterization were performed on the day of ink preparation and every day up to a week after preparation to examine the stability of the ink. The ink was stored at room temperature (RT) and in a fridge (FT) to analyze the storing conditions’ effect on the stability.

#### 3.2.1. Rheology of the Printing Ink

Rheology measurements were performed in the same way as in the aforementioned study by Sjöholm et al. [26]. In the current study, the aim was to investigate the viscosity of both the drug-loaded printing ink and the drug-free placebo solution under the shear rate for 0–7 days in order to study the stability of the ink and possible change in viscosity over time, which could have an effect on the printing results. The TPH-containing printing ink prepared in this study was also compared to the results obtained for the GBP-containing ink in the previous study [26].

#### 3.2.2. Solid-State Characterization of the Printing Ink

In attenuated total reflection–Fourier transform infrared spectroscopy (ATR-FTIR), a solid sample is placed on an ATR crystal with a high refractive index. Infrared radiation is sent through the crystal and is totally reflected at the surface between the two different optical media. However, a small fraction of the radiation will also extend into the sample, where it is absorbed to various extents based on the composition of the sample. Therefore, the totally reflected infrared radiation is slightly attenuated since it lacks the absorbed parts. The measured data is translated into a spectrum [31]. 

Throughout the stability study of the printing ink, the solid-state was investigated with ATR-FTIR spectroscopy (UATR-2 Spectrum Two, PerkinElmer, Llantrisant, UK) over a range of 4000 to 400 cm^−1^ with 4 accumulations at a resolution of 4 cm^−1^ and analyzed with PerkinElmer software Spectrum (v. 10.03.02). A force of 75 N was applied onto the sample while measuring, and the spectrum was further treated with the program functions of baseline correction, normalization (3% T), and data tune-up. Each sample was measured in duplicate. A third measurement was performed in case of inconsistencies.

### 3.3. Computer-Aided Design

In the previous study [26], seven different sizes were designed and printed to show the printer’s capability to produce tailored doses. The correlation between designed size and obtained dry weight and between obtained wet weight and obtained dry weight were R^2^ = 0.9966 and R^2^ = 0.9998, respectively [26]. The study demonstrated that the prepared formulation and the printing technology were suitable for producing tailored drug doses. In the current study, four different sizes were considered sufficient to show the correlation between the pre-determined design and the obtained drug amount in the printed dosage form. The four different sized rectangles were designed utilizing a computer-aided design (CAD) software (Autodesk Fusion 360 by Autodesk, San Rafael, CA, USA, 2.0.10446, 2020). The designed rectangles had the same width and height of 10 and 1 mm, respectively, but they were designed with increasing lengths (5, 10, 15, and 20 mm) and were named according to their length.

### 3.4. Semi-Solid Extrusion 3D Printing

The Brinter 1 3D BioPrinter (Brinter Ltd., Turku, Finland), a semi-solid extrusion (SSE) 3D printer connected with a Pneuma Tool print head (Brinter Ltd., Turku, Finland) and a compressor, was successfully used in the previous study [26], and hence was the printer choice in the current study as well. The pre-loaded syringes attached with 16 G precision tips (Nordson EFD LLC, East Providence, RI, USA) were placed into the Pneuma Tool print head prior to printing. In the browser-based Brinter printing software, the print settings were set to the same as in the previous study, i.e., 1 mm layer height, one shell, and solid infill with a 45-degree fill angle. The print speed was set to 8 mm/s and the pressure to 1500 mbar during printing to achieve therapeutic doses. All parameters, except the printing pressure, were kept the same as in the previous study. The printed dosage forms were printed on transparent poly sheets (Folex imaging X-10.0 copier films, Paper Spectrum Limited, Leicester, UK) placed on an analytical balance for quality control. The printed ChewTs were left to dry at room temperature overnight prior to analysis.

### 3.5. Characterization of the Dosage Forms

#### 3.5.1. Physical Appearance

The prepared printed ChewTs were examined visually and photographed. The weights of the printed dosage forms were determined by an analytical balance (Radwag Wagi Elektroniczne, Radwag, Radom, Poland), and the thickness was determined by measuring the middle and all four corners of the dosage form with a caliper (Absolute Digimatic by Mitutoyo Corp., Kawasaki, Japan). Average and standard deviations for the different sized printed ChewTs were calculated (n = 5). 

#### 3.5.2. Drug Content

Accurate and precise drug content is of the highest importance in the production of drug dosage forms, especially for tailored doses. In the current study, four different sizes were printed to investigate the printer’s ability to produce drug doses with high precision, namely, that increased drug content can be achieved in correlation with increased size. Five dosage forms of each size were measured to determine the drug content of the printed ChewTs. Each ChewT was dissolved in sealed 250 mL borosilicate flasks containing 100 mL of purified water prior to being fixed and shaken on an orbital shaker (Multi-Shaker PSU 20 by BIOSAN, Riga, Latvia) at 150 rpm for three hours. The samples were appropriately diluted, and the absorbance was spectrophotometrically measured at 272 nm with a UV-6300PC Double Beam Spectrophotometer (VWR International BVBA, Leuven, Belgium). The spectrophotometer was equipped with matching 10 mm quartz cells (QS High Precision Cell, Hellma Analytics, Müllheim, Germany). Data were gathered and analyzed with the UV-Vis Analyst software v. 5.44 (VWR International BVBA, Leuven, Belgium). The system was zero-calibrated before measurement with corresponding drug-free placebo samples treated in the same manner as the drug-loaded samples in order to omit any potential absorbance from the excipients.

To investigate the accuracy of the printing technology, the uniformity of dosage units was determined. The uniformity of dosage units can be demonstrated by two methods, namely mass variation (MV) and content uniformity (CU). The test for MV is only applicable on solid dosage forms that contain 25 mg or more of an active ingredient compromising of 25% or more, by mass, of the dosage unit. The CU test is applicable in all cases. Both tests were applicable in our study, but only CU was determined. The European Pharmacopoeia (Ph. Eur.) specifies under chapter 2.9.40. [32] that no less than 10 dosage units should be investigated. In the present study, five units each of the four different-sized printed ChewTs were measured. The acceptance value (AV) is calculated and the requirements for dosage uniformity are met if the AV of the first 10 dosage units is less or equal to 15. In case a greater AV is obtained, 20 more dosage units should be measured, and the requirements are met if the final AV of the 30 dosage units is less or equal to 15 and no individual content of the dosage units is outside the criteria given in the said chapter. The United States Pharmacopoeia (USP) uniformity of dosage units is harmonized with the Ph. Eur. uniformity of dosage units.

#### 3.5.3. Salivary pH

As these printed dosage forms are expected to be chewed by the animals, the salivary pH of the final printed ChewTs was measured. It is widely accepted that the salivary pH of chewable dosage forms should be neutral or close to 7 to avoid oral irritation [33]. Determination of the salivary pH has been performed in various ways, but the general principle is to wet a piece of the dosage form in a small amount of water and measure the pH of the liquid. In this study, the method previously reported by Sjöholm et al. [34] was chosen. The measurement was performed on size 10 printed ChewTs (n = 3) at room temperature with an electronic pH meter (Edge R pH HANNA Instruments, Inc., Woonsocket, RI, USA). The ChewT was wetted with 1 mL of sterile water in a glass vial upon after 30 s, the pH electrode was brought to the water surface. Equilibrium was allowed for 1 min prior to recording the reading. The average values with standard deviations were calculated from the pH and temperature readings.

#### 3.5.4. In Vitro Disintegration

Rapid disintegration is especially beneficial for veterinary dosage forms, as fast disintegration will prevent the dosage form from being spit out or getting stuck in the esophagus. In the present study, the disintegration test was conducted in the same manner as performed in the previous study [26], where a thorough description of the method can be found, to investigate the disintegration time of the TPH-containing size 10 printed ChewTs (n = 6). The printed ChewTs disintegration was observed throughout the measurement, and the time to complete disintegration was recorded.

#### 3.5.5. In Vitro Dissolution

The drug release studies were carried out on the pure TPH and TPH-containing ChewTs of size 5. The drug release study was performed in accordance with the European Pharmacopoeia Section 2.9.3, dissolution test for solid dosage forms [35]. The test was performed utilizing automatic dissolution (Sotax AT 7 Smart, Basel, Switzerland). The thickness and weight of each ChewT were noted prior to being placed in the vessel containing 900 mL of purified water. The ChewTs were placed into spiral sinkers to prevent them from floating. Rotating paddles were attached, and the dissolution system was set to have a speed and a temperature of 100 rpm and 37 °C, respectively. Aliquots of release media were automatically withdrawn at pre-determined time points using an automatic pump system (Sotax CY 6, Basel, Switzerland). These automatically withdrawn samples were filtered through glass microfiber filters GF/B (GE Healthcare Life Sciences, Cheshire, UK) prior to UV-Vis spectrophotometric measurement (Lambda 35, PerkinElmer, Singapore) at a wavelength of 272 nm. The average and standard deviations of the released drug amounts were calculated from the obtained data for the pure TPH and TPH-containing printed dosage forms (n = 3). The dissolution profiles of percentual drug release were plotted against time.

### 3.6. Stability Study

In the current study, the authors decided to study the stability of the prepared ChewTs as they expect the pet owners to store them at home. Printed ChewTs of size 10 were hence stored at room temperature in closed Petri dishes. The size 10 printed ChewTs were analyzed on days 1, 3, 7, 14, and 21, and on 1, 2, and 3 months after printing. On the different stability study time points, drug content, moisture content, mechanical strength, and solid-state characterization were performed.

#### 3.6.1. Drug Content

The drug content of size 10 printed ChewTs was measured, and the CU was determined, both according to the methods described in Section 3.5.2, on the time points mentioned in Section 3.6.

#### 3.6.2. Moisture Content

Moisture content can be a crucial attribute of the final dosage form, as a too low or too high moisture content may influence the chemical and physical stability of the final product. The moisture content of the printed ChewTs was measured with a moisture analyzer (Radwag Mac 50/NH by Radwag, Radom, Poland) in triplicate on the said stability study time points. The weight loss in mass-%, which is equal to the moisture content, was recorded by heating up the samples to 120 °C and recording the endpoint of the test where the change in mass was less than 1 mg/min. The average values with standard deviations (n = 3) were calculated, and the room temperature and relative humidity were monitored during the tests.

#### 3.6.3. Mechanical Strength

To ensure the ability to pack, handle, and administer the dosage form, sufficient mechanical strength is required. The measurement was conducted in the same manner as the previous study conducted by Sjöholm et al. [26], with the exception of a cylindrical probe utilized instead of a ball probe. To investigate the hardness of the TPH-containing ChewTs, printed ChewTs of size 10 were analyzed (n = 5). The average values with standard deviations were calculated for the crushing point (N) and the travel distance (mm). The temperature and relative humidity were monitored and recorded during the analysis. The test was performed at every stability study time point in order to investigate changes in the hardness over time. 

#### 3.6.4. Differential Scanning Calorimetry

Differential scanning calorimetry (DSC) is a technique that measures the energy required to increase the temperature of a sample. The heat flow to and from a sample is obtained as a function of temperature [36]. Exothermic events such as crystallization release heat from a sample, while endothermic events such as evaporation, melting, and glass transition take heat into the sample. Thus, events such as the dehydration point, melting point, and glass transition temperature can be observed as peaks in the DSC thermogram. Hence, the purity, polymorphism, stability, and interactions can be investigated [37]. 

In the current study, the thermal properties of the printed drug-loaded dosage forms, drug-free placebo extrudates, raw materials, and physical mixtures were determined with DSC (Q2000 instrument by TA Instruments, New Castle, DE, USA) and analyzed with the TA Universal Analysis software v. 4.5 A by TA Instruments. The samples of approximately 3 mg were weighed and sealed in Tzero aluminum pans with matching Tzero lids (TA instruments, Switzerland) and measured through a heating ramp from 20 to 300 °C with a heating rate of 10 °C/min under nitrogen purge gas with a flow rate of 50 mL/min. A minimum of two measurements were carried out for each sample, and if any differences were observed, a third measurement was performed. 

#### 3.6.5. Attenuated Total Reflection–Fourier Transform Infrared Spectroscopy

The solid-state of the pure substances, physical mixtures, printed dosage forms, and drug-free placebos were investigated with ATR-FTIR spectroscopy in accordance with the method described in Section 3.2.2. 

#### 3.6.6. Raman Spectroscopy

Raman spectroscopy is a non-destructive light scattering chemical analysis technique that provides information about the molecular interactions of materials, chemical structure, polymorphic forms, and crystallinity [38]. In this study, the polymorphic forms of TPH were investigated by Raman spectroscopy. The pure substances, physical mixtures, drug-loaded printed ChewTs, and drug-free placebo extrudates were analyzed with Raman spectroscopy Nicolet iS50 Raman by Thermo Fisher Scientific (Waltham, MA, USA) equipped with a 1064 nm diode laser (500 mW) and an InGaAs detector. Using the 1064 nm laser has been found the most appropriate when measuring drugs with Raman spectroscopy [39]. Approximately 3 mg of sample was placed on a gold plate, focused through OMNIC µView (v. 9.1.0) software prior to measurement, and measured with 128 scans at 0.25 W and the aperture set to 50. Spectra were acquired using OMNIC iS50 Raman software (v. 9.1.0) at a spectral range of 3600–100 cm^−1^. Two measurements were performed on each sample. 

## 4. Results and Discussion

### 4.1. Ink Characterization

The animals’ voluntary uptake of the medicinal products is desirable, and hence the palatability plays a crucial role [40]. Therefore, taste-enhancers are commonly added to the formulation in order to increase voluntary uptake induced by taste and smell. In the current solution, liver powder was added as a taste-enhancer which yielded brown-colored solutions with an apparent scent of liver. Both the drug-loaded and drug-free placebo printing inks were visually inspected to confirm the homogeneity of the printing solution. A bubble-free drug-loaded ink suitable for SSE 3D printing was obtained. The placebo solution exhibited a lower viscosity than the drug-loaded solution and could not keep shape upon printing and was hence manually extruded for the required analysis.

### 4.2. Rheology

The rheological properties of the drug-loaded printing ink and the drug-free placebo solution stored at room temperature and fridge temperature were examined against shear rate once a day for up to a week (Figure 1). As observed in the figure, the printing inks stored at different temperatures and measured at different time points all exhibited a shear-thinning behavior. The drug-loaded inks exhibited higher viscosities compared to the drug-free placebo solutions. The intermolecular interactions and increased volume fraction will generally occur between the drug and other excipients in the formulation [41]. The increased viscosity observed in the drug-loaded solutions is due to the addition of TPH, which increases the solid fraction in the solutions. This was also observed in the aforementioned study containing GBP [26].

In the current study, the stability of the printing solution was investigated, and an increase in viscosity for each day was observed. Still, no difference between the two storing temperatures was noticed. This increase in viscosity could affect the printing result, and further investigation is needed. A correlative study should be conducted, where the time from preparation and the pressure required to reach therapeutic doses would be investigated. Increased pressure is required to achieve the same print result with higher viscosity solutions. This study performed the printing on the same day of preparation (day 0).

When comparing the rheology results obtained in the current study as opposed to the results obtained in our previous study containing the same ink base [26], the current study’s TPH-containing printing ink had a higher viscosity. The higher viscosity of the solution might be associated with the molecular weight and particle size of the API. TPH has a molecular weight of 180.16 g/mol and a particle size of >500 microns in diameter [42], which is higher than the molecular weight and particle size of GBP, which are 171.24 g/mol and 28.8 µm, respectively. However, more importantly, GBP is freely soluble in water, whereas TPH is only slightly soluble in water [43,44], giving rise to a higher viscosity of the formulation. Therefore, due to the higher viscosity exhibited by the ink used in the current study, it was expected that higher pressure would be required to be applied to obtain therapeutic doses.

### 4.3. Solid-State Characterization of the Printing Ink

ATR-FTIR was performed on printing inks stored at different temperatures over a range of 4000 to 400 cm^−1^ with 4 accumulations at a resolution of 4 cm^−1^. The ATR-FTIR results of the pure TPH, TPH-loaded printing inks stored at room temperature (RT) and in a fridge (FT) throughout the stability study, and the corresponding drug-free placebo solution analyzed on the day of preparation are all presented in Figure 2 in a range of 1800 to 600 cm^−1^. The spectra were treated with baseline correction, normalization (3% T), and data tune-up.

TPH is known to exist as a crystalline monohydrate and three anhydrous polymorphs (the stable forms I and II and the metastable form III). A fourth anhydrous form IV has also been identified. Both the monohydrate and the anhydrous form II are commercially available, but the anhydrous form is more commonly used as an API in pharmaceutical dosage forms. TPH is known to undergo phase transition during pharmaceutical processing. When placed in contact with water at ambient temperature, anhydrous TPH transforms into TPH monohydrate [3,42].

In Figure 2, the FTIR absorptions bands found at 1705 and 1665 cm^−1^ are assigned to the C=O stretching vibration of the carbonyl groups, and the band at 1559 cm^−1^ is attributed to the stretching vibration of the imine group of anhydrous TPH [45]. The TPH hydrate exhibits O-H vibrational stretching at 3360 cm^−1^, N-H vibrational stretching at 3131 cm^−1^, C-H vibrational stretching at 3106 cm^−1^, and C=O vibrational stretching at 1695 and 1642 cm^−1^ [46]. None of the aforementioned absorption bands were found in the prepared dosage forms, indicating the absence of TPH hydrate. The characteristic peaks of forms I and II are very similar, except that TPH form I is known to exhibit a characteristic peak at 1148 cm^−1^. The lack of this characteristic peak in the prepared drug-loaded printing ink indicates that polymorphism is not found in the sample [47]. From the FTIR results, the conclusion can be drawn that the prepared drug-loaded printing ink contains anhydrous TPH of polymorph II. No changes to the drug were observed by the preparation method nor throughout the stability study of the ink. This indicates that we have the active and stable form of TPH in our prepared printing ink throughout the one-week stability study on the ink. No differences between the inks stored at different temperatures were noticed, indicating that the drug stays intact in both storing conditions. 

It is worth mentioning that microbial growth was not investigated, and as the ink contains water, this would further be needed to be explored. Furthermore, while printing, the temperature affects the viscosity of a paste or a gel, and the printing should always be performed at a similar temperature; hence if the ink is stored at lower temperatures, the ink needs to be allowed to reach room temperature prior to printing or alternatively a temperature-controlled printer should be utilized.

### 4.4. Semi-Solid Extrusion 3D Printing

TPH-containing ChewTs were successfully printed utilizing the Brinter printer. The prepared drug-loaded printing ink was printable and a great correlation between the designed sizes and obtained wet weights of the printed ChewTs was achieved (R^2^ = 0.9972). The optimal print settings found in the previous study [26] were also suitable for the current study. As observed in the rheological results between the two studies, the TPH ink exhibited a higher viscosity, and the conclusion was drawn that a higher printing pressure would be required. As expected, a higher pressure was required, as the pressure applied for printing therapeutic doses of TPH was 1500 mbar, compared to 290 mbar for the GBP ink, all other parameters were kept constant. 

### 4.5. Physical Appearance

In Figure 3, the obtained brown-colored, rectangular-shaped printed ChewTs with escalating doses can be seen. As in the previous study, an increasing thickness (from 1.47 to 1.68 mm) with increasing size was similarly observed in the current study. This is due to the ink’s surface-tension-dominating behavior, which also causes the corners of the printed ChewTs to round. The physical characteristic, i.e., the product’s mouthfeel, is essential to achieve voluntary acceptance by the animals. It has been found that animals prefer harder chewable dosage forms resembling an animal treat compared to humans, that tend to prefer a chewier dosage form [40]. The prepared ChewTs exhibited a similar texture to a dog or a cat treat, with an apparent liver scent which the authors expect to be favorable for the animals and is assumed to result in high voluntary acceptance but would remain to be tested and proven.

### 4.6. Drug Content

In order to produce tailored drug doses, a high correlation between designed size and obtained drug amount is vital. Hence, the drug content was measured for all four doses (n = 5). Currently, marketed TPH products contain excessive doses to be administered to animals. The aim was, therefore, to successfully produce lower doses of TPH. The prepared ChewTs contained 30–120 mg TPH with a high correlation between the designed size and the obtained drug amount (R^2^ = 0.9973). These doses are suitable for a dog weighing 10 kg and above. Appropriate doses for cats and smaller dogs could easily be achieved by decreasing the drug concentration in the printing ink. 

The CU for the four sizes was determined. The calculation should be performed using 10 units, but only 5 units were measured in the current study. Still, the acceptability constant for 10 doses was employed, *k* = 2.4 (as there is no set acceptability constant for using 5 doses). The requirements are met if the CU is less or equal to 15. The calculated CUs for the prepared ChewTs were 6.78, 13.75, 8.62, and 8.48 for sizes 5, 10, 15, and 20, respectively. All sizes met the Ph. Eur. and USP criteria for uniformity of dosage units [32]. This confirmed that SSE 3D printing technology could be utilized for the production of tailored doses close to the point-of-care.

### 4.7. Salivary pH Test

The obtained pH for the TPH-loaded ChewTs was 6.3 ± 0.0, which is slightly more acidic than the obtained pH for the GBP-loaded ChewTs with an obtained value of 7.0 ± 0.4 [26]. In a study performed by Patel et al., dosage forms with surface a pH of 4.5–6.5 did not show any local irritation [48]. Hence, we expect that the prepared dosage forms will be non-irritable and should not cause any discomfort upon administration. 

### 4.8. In Vitro Disintegration

According to the FDA guideline, the disintegration time of chewable dosage forms should be short enough to prevent gastrointestinal obstruction if the patient does not completely chew the dosage form. Still, no specific time limit is stated [49]. Nyamweya et al. studied the disintegration time of several commercially available chewable tablets in a similar setup as used in the current study. They found that the mean disintegration times ranged from 6 min to more than 60 min in distilled water [50]. The TPH-loaded ChewTs prepared in the current study disintegrated at 7 min and 47 s with a standard deviation of 39 s. The obtained result is in the lower end of the spectrum found by Nyamweya et al., but more than double the time it took for the GBP-loaded ChewTs to disintegrate in the previous study [26]. The poorer solubility of TPH compared to GBP might attribute to the difference in the disintegration time between the two printed ChewTs. However, the disintegration time of the prepared dosage forms in both studies is lower than that of commercially available chewable dosage forms found by Nyamweya et al. [50] and are hence considered sufficient.

### 4.9. In Vitro Dissolution

The dissolution profiles of the pure drug and the drug-loaded ChewTs, plotted as cumulative drug release vs. time, are shown in Figure 4. Pure theophylline achieves 100% drug release within minutes. The drug release from the printed dosage form is slightly slower, with 100% drug release at 5 min. The drug release between the TPH dosage forms and the GBP dosage forms cannot be compared as the dissolution setups are different and hence not comparable. However, both dosage forms exhibit rapid drug release appropriate for a chewable dosage form.

### 4.10. Stability Study

#### 4.10.1. Drug Content

The drug content of the stored ChewTs was determined on each of the stability study time points. The ChewTs’ measured dry weight, obtained drug amounts, and the calculated CUs, with corresponding reported temperature and humidity on the different stability study time points, are displayed in Table 1. The dosage forms were designed to contain 60 mg of TPH, and as can be seen in the table, the obtained drug amount was slightly higher, with an average drug content of 62.7 mg for all different time points combined. A slight downward trend can be observed during the stability study. This could be due to a slight decrease in the API activity or small differences in the printing result. However, no significant changes in drug content during the three-month stability study were observed. More importantly, the uniformity of dosage form (≥15) criteria was met on all stability study time points. 

#### 4.10.2. Moisture Content

The moisture content present in a pharmaceutical dosage form can have an effect on the chemical, physico-chemical, and microbial stability of the product. In addition to the stability of the product, moisture content can also affect handleability. For example, too low moisture content may yield hard and brittle tablets, while a too high moisture content might instead yield soft and brittle tablets. The suitable moisture content of a tablet was found by Tomar et al. to be around 4–5% [51]. In the current study, the average measured moisture content at the different stability study time points varied between 3.2% and 4.7%, which is in line with the study found by Tomar et al. This is also in line with the previous study’s findings [26].

#### 4.10.3. Mechanical Strength

The crushing strength and distance of travel measured at the different time points during the stability study are visualized in Figure 5. The formulation contains HPMC, which is a hygroscopic polymer, i.e., it absorbs moisture from the ambient environment [52], this generally causes a dosage form to become weaker [51]. In the current study, this was observed. A reversed correlation between the recorded humidity on the study time point (Table 1) and the corresponding crushing strength and distance of travel for the dosage form (Figure 5) can be seen, with a correlation coefficient of −0.83. In other words, a higher humidity caused a loss in strength. The average crushing strength for each time point varied between 20 and 45 N and the distance of travel between 0.4 and 0.6 mm, both being highest on Day 1. The TPH-containing ChewTs required a much higher crushing strength compared to the GBP-containing ChewTs that required a crushing strength of only around 10 N [26], the results can not directly be compared as a different probe was utilized for the said measurements. The study by Nyamweya et al. [50] also studied the tablet breaking force of various commercially available chewable tablets with a hardness tester. They obtained values between 70 and 150 N. In order to compare results from different studies properly, the setup needs to be identical. This, unfortunately, was not the case between the current study and the study by Nyamweya et al., where they utilized a hardness tester instead of a texture analyzer. The use of a hardness tester was also attempted in the current study, but the test was not applicable due to the shape of the dosage form. The FDA recommends a chewable tablet having a hardness of less than 12 kp [49], which corresponds to 118 N. The ChewTs prepared in the current study are in line with this.

#### 4.10.4. Differential Scanning Calorimetry

DSC was performed on pure substances, physical mixtures, drug-loaded printed ChewTs, and drug-free placebo extrudate at a temperature range between 20 and 300 °C. The DSC results of the pure TPH, TPH-loaded ChewTs measured during the different time points of the stability study, and the corresponding drug-free placebo extrudate measured on day 1 of the stability study in a temperature range from 40 to 280 °C are presented in Figure 6. 

The melting point of form II has been reported in the literature at 269.1 ± 0.4 °C by Suzuki et al. [53] and at around 270 °C by Zhu et al. [47]. The anhydrous form I has a higher melting temperature at about 275 °C [47,54]. In the current study, pure TPH exhibits an endothermic peak at 271.81 °C, with a *T*_onset_ of 269.83 °C corresponding to a melting point of anhydrous TPH form II. In the drug-loaded printed ChewTs and the drug-free placebo extrudate, only one endothermic peak at around 154 °C is observed, which is attributed to a shifted peak corresponding to mannitol in the formulation. The previous study also observed that the same ink base provided an endothermic event in the same region [26]. The drug-loaded ChewTs did not show an endothermic event corresponding to the melting of TPH. This could indicate that the drug is in an amorphous state, but it is unlikely due to the observation of drug crystals in the prepared dosage forms. No additional peaks were observed throughout the stability study. Ultimately, DSC could not be used to analyze the polymorph in the final dosage form.

#### 4.10.5. Attenuated Total Reflection–Fourier Transform Infrared Spectroscopy

ATR-FTIR was performed on pure substances, physical mixtures, drug-loaded printed ChewTs, and drug-free placebo extrudate over a range of 4000 to 400 cm^−1^ with 4 accumulations at a resolution of 4 cm^−1^. The ATR-FTIR results of the pure TPH, TPH-loaded ChewTs measured during the different time points of the stability study and the corresponding drug-free placebo extrudate measure on day 1 of the stability study in a range of 1800 to 600 cm^−1^ are presented in Figure 7. The spectra were treated with baseline correction, normalization (3% T), and data tune-up. 

The same absorbance bands found in the printing ink in Figure 2, described in Section 4.3, are also observed in the printed ChewTs in Figure 7. Therefore, the conclusion can be drawn that the prepared ChewTs contain anhydrous TPH of polymorph II. Furthermore, no changes to the drug were observed by the preparation method nor throughout the stability study. This indicates that we had the active and stable form of TPH in our prepared dosage forms as well as the stability of the drug in the ChewTs over a three-month period.

#### 4.10.6. Raman Spectroscopy

Raman spectroscopy was used to investigate the polymorphic character of TPH in the prepared ChewTs and whether any change was observed over three months. Measurements were performed on pure substances, physical mixtures, drug-loaded printed ChewTs, and drug-free placebo extrudate at a spectral range of 3600 to 100 cm^−1^. The Raman results of the pure TPH, TPH-loaded ChewTs measured during the different time points of the stability study, and the corresponding drug-free placebo extrudate measured on day 1 of the stability study are presented in a range of 1800 to 400 cm^−1^ in Figure 8. 

The characteristic peaks of TPH are highlighted in the figure and are all in line with the literature [55]. The bending vibrations of C-N-C can be found at 449 cm^−1^, N-C-C at 499 cm^−1^, and C=C-C at 550 cm^−1^. At 925 and 966 cm^−1^, the bending vibrations of N-C-H and N=C-H, respectively, can be found. The vibrational stretching of C-C can be observed at 1186 cm^−1^ and the vibrational stretching of C-N bands is found at 1060, 1243, 1283, and 1311 cm^−1^. At 1429 cm^−1^, the bending vibration of H-C-H can be detected. Peaks found at 1564, 1607, and 1703 cm^−1^ correspond to the vibrational stretching of C=O, C=C, and C=N, respectively. All the mentioned peaks of TPH are observed in the spectra of the pure drug and in all the measured time points of the drug-loaded ChewTs, and no variation throughout the stability study could be observed in the prepared dosage form. The absence of a peak at 1329 cm^−1^, characteristic to polymorph I, and the presence of the vibrational stretching of C=O at 1662 cm^−1^, characteristic to polymorph II in the prepared samples [47], confirms the results found in the FTIR spectra that anhydrous TPH of polymorph II is present in the printed dosage form and that the formulation did not change the TPH polymorph nor were any changes observed over the three-month period.

## 5. Conclusions

The present study aimed to investigate semi-solid extrusion (SSE) 3D printing as a novel automated approach to substitute the current labor-intensive manual compounding practice to answer the unmet need for available and suitable veterinary products. The current study demonstrated that the same ink base utilized in a different study was also appropriate for preparing theophylline (TPH)-containing dosage forms. Four different doses were successfully printed with a high correlation between designed size and obtained drug amount, proving the technique’s suitability to produce tailored doses. The stability study showed that the prepared TPH-loaded chewable tablets (ChewTs) with the addition of liver powder for higher acceptance were stable to use throughout the study, namely, for three months. The findings in this study carry the practical benefit of demonstrating that pet-friendly ChewTs with precise TPH doses can easily be manufactured close to the point-of-care. The results gained can give direction to other areas where dose tailoring is required, such as pediatrics.

## Figures and Tables

**Figure 1 pharmaceutics-14-01339-f001:**
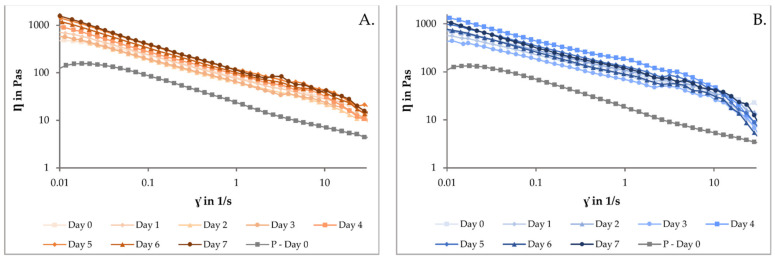
Viscosity vs. shear rate curves was measured on the day of preparation and daily for a week after preparation. Solutions stored at room temperature are visualized in orange (**A**), and solutions stored at fridge temperature are visualized in blue (**B**). Corresponding drug-free placebo solution was measured only on the day of preparation; it is visualized in gray in both figures and marked with “P” for placebo.

**Figure 2 pharmaceutics-14-01339-f002:**
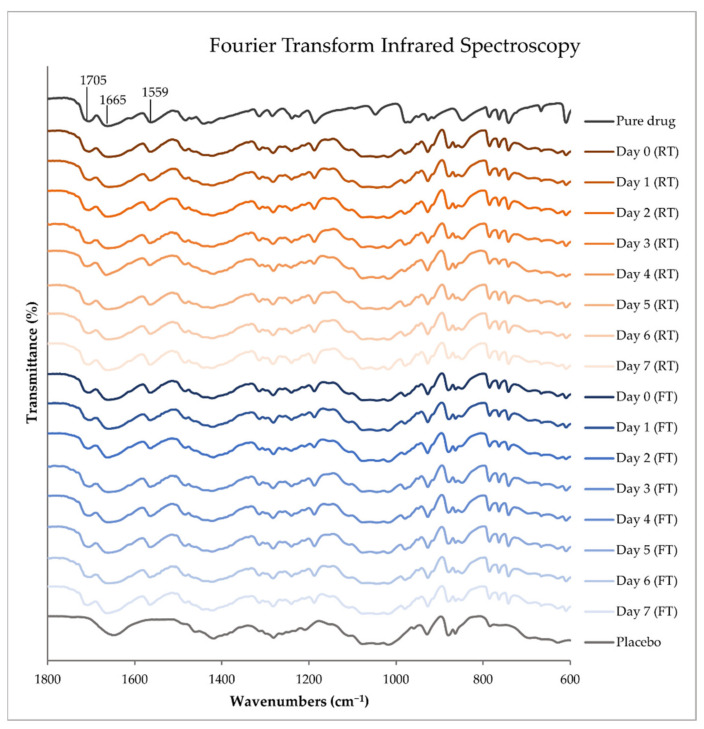
ATR-FTIR spectra of pure theophylline and drug-free placebo solution were analyzed on the day of ink preparation. The drug-loaded printing inks were stored at room temperature (RT) and in a fridge (FT) and were analyzed on the day of preparation and up to one week after preparation.

**Figure 3 pharmaceutics-14-01339-f003:**
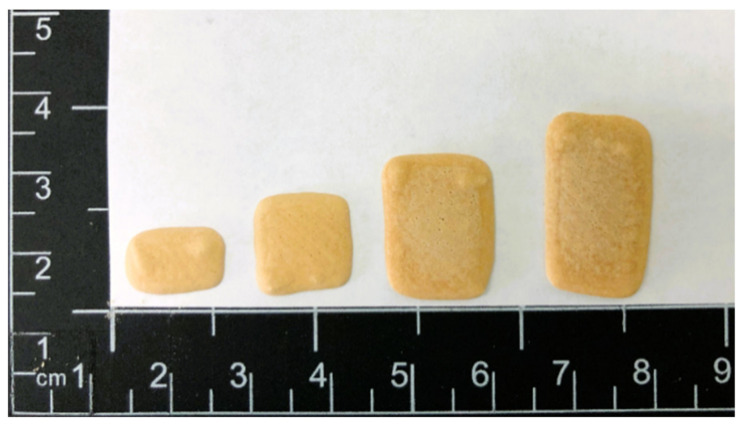
A photographic picture of different sized semi-solid extrusion 3D-printed theophylline-containing ChewTs.

**Figure 4 pharmaceutics-14-01339-f004:**
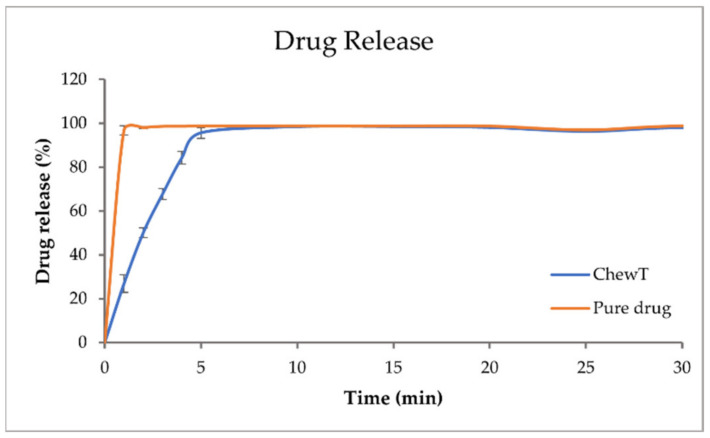
The drug release profile of pure theophylline and theophylline-loaded printed ChewTs in purified water (n = 3).

**Figure 5 pharmaceutics-14-01339-f005:**
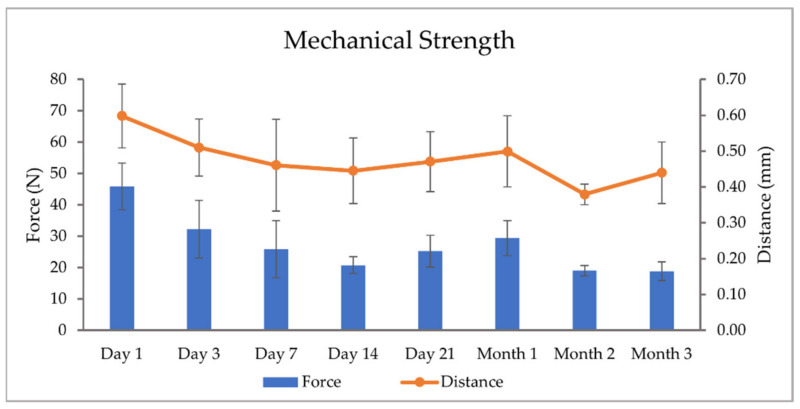
Measured crushing strengths (N) and distance to break (mm) of the ChewTs on the different stability study time points (n = 5).

**Figure 6 pharmaceutics-14-01339-f006:**
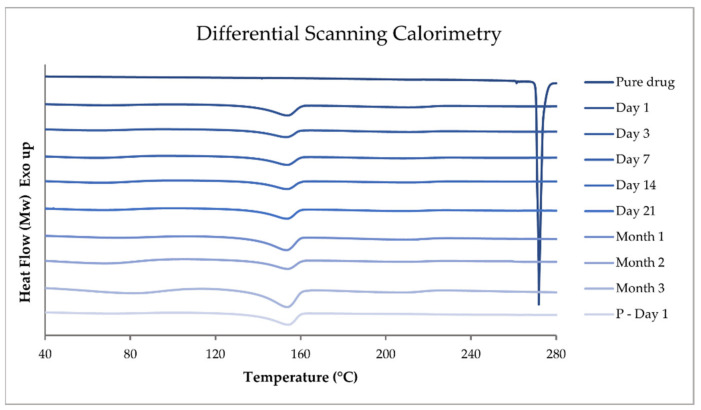
DSC endothermic curves of pure theophylline and drug-loaded printed ChewTs were measured at the different stability study time points. In addition, drug-free placebo (P) extrudate was measured on day 1 of the stability study.

**Figure 7 pharmaceutics-14-01339-f007:**
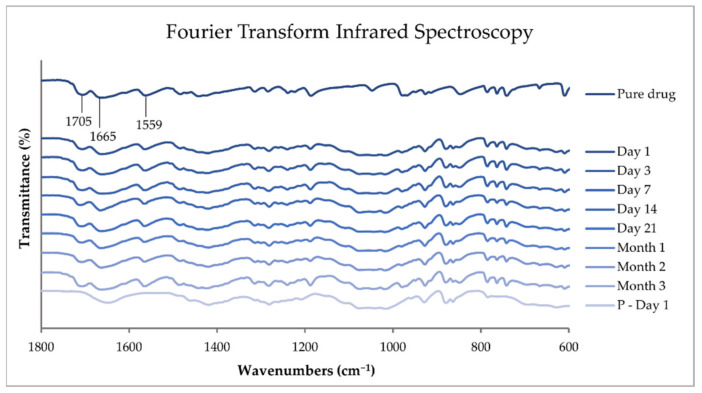
ATR-FTIR spectra of pure theophylline and drug-loaded printed ChewTs were measured at the different stability study time points, and drug-free placebo (P) extrudate was measured on day 1 of the stability study.

**Figure 8 pharmaceutics-14-01339-f008:**
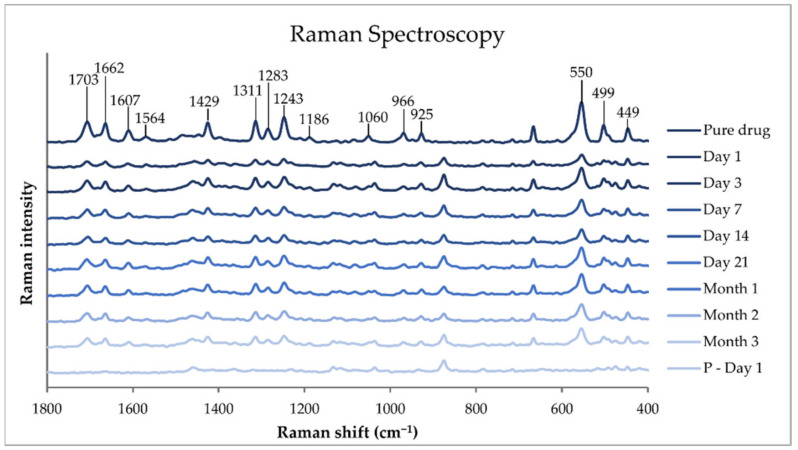
Raman spectra of pure theophylline and drug-loaded printed ChewTs were measured at the different stability study time points, and drug-free placebo (P) extrudate was measured on day 1 of the stability study.

**Table 1 pharmaceutics-14-01339-t001:** Measured average drug content and dry weight with standard deviations and calculated content uniformity (CU) on the pre-determined stability study time points (n = 5). Reported is also the recorded average relative humidity and temperature on each measurement day.

Stability Study Timepoint	Dry Weight (mg)	Drug Amount (mg)	Content Uniformity	Humidity (RH%)	Temperature (°C)
Day 1	205.6 ± 4.8	64.3 ± 2.0	13.75	22.1 ± 0.1	21.4 ± 0.0
Day 3	198.8 ± 5.6	64.7 ± 1.5	12.22	31.6 ± 0.1	21.5 ± 0.0
Day 7	198.1 ± 8.1	62.9 ± 1.4	9.05	38.2 ± 0.1	21.9 ± 0.0
Day 14	207.8 ± 7.0	66.1 ± 1.2	13.32	39.9 ± 0.1	21.1 ± 0.1
Day 21	199.7 ± 4.8	61.5 ± 1.6	7.52	29.7 ± 0.2	21.4 ± 0.1
Month 1	194.7 ± 3.1	60.7 ± 0.8	3.27	35.2 ± 0.1	22.4 ± 0.0
Month 2	204.9 ± 9.5	60.6 ± 2.5	10.12	55.0 ± 0.3	22.1 ± 0.1
Month 3	208.5 ± 5.3	61.0 ± 1.0	4.00	42.5 ± 0.2	21.7 ± 0.0

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
