# Peer review of "Compounding Tailored Veterinary Chewable Tablets Close to the Point-of-Care by Means of 3D Printing"

_pharmaceutics, 2022, doi:10.3390/pharmaceutics14071339_

Round 1
Reviewer 1 Report
The authors of this work 3D printed personalised dosage forms for dogs using semi-solid extrusion (SSE). The ink they have used is the same that the authors used in a previous work, also for veterinary applications, but the drug gabapentin used in the previous work has been replaced by theophylline.
The current study aimed to examine if the same printing ink base used in the gabapentin study could be utilized with a different drug to simplify the SSE 3D printing by providing one base suitable for preparation of several veterinary drug products. The authors also investigate the drug's influence on the dosage form. the authors should update the bibliography with more recent papers on SSE extrusion which have been published in the last few years
They performed rheology studies and solid-state experiments (FTIR) with the ink and characterised the dosage forms in terms of drug content, in vitro disintegration and dissolution studies, DSC and Raman among others. In general, the characterization of the ink and printed tablets has been properly done.
The authors considered the problem of voluntary uptake of the medicinal products by animals, which is highly relevant to the success of a veterinary dosage form. They added taste-enhancers (liver powder) to the formulation to increase voluntary uptake induced by taste and smell.
When they compared the rheology results obtained in the current study as opposed to the results obtained in the previous study containing the same ink base, the theophylline-containing ink showed a higher viscosity, which might be associated with the higher molecular weight and particle size of theophylline compared to gabapentin, but they also associated it with the poor water solubility of theophylline with leads to a higher viscosity of the ink. They also link this poor water solubility to the longer disintegration time showed by the theophylline printlets. This increase in ink viscosity may require some printing parameters to be changed, but the authors did not state if they changed any parameters in the 3D printer compared to the previous study.
Stability studies up to three months were also conducted, showing no significant variations in the drug content of the printlets, which is positive.
In general, the experiments are properly done, and the results well explained. In terms of originality, it does not add anything new because they reuse an ink that they had already developed in a previous work, and only change the drug. In addition, they use the same printing technique, and their dosage forms have the same application as in their previous work. However, their concept of developing an ink that can be used for any drug could be interesting.
Author Response
Response to Reviewer 1 Comments
The authors appreciate the comments made by the reviewer and the opportunity to improve the manuscript. The responses to the points brought up by the reviewer are provided below.
Point 1: The authors should update the bibliography with more recent papers on SSE extrusion which have been published in the last few years.
Response 1: We agree with the reviewer, and we have now included published papers from 2020 to date where SSE 3D printing was explored for the production of various drug delivery systems. The following text has been added to the manuscript on rows 83-90 "Lately, SSE 3D printing has gained interest as a manufacturing technique for the production of drug-loaded dosage forms. In addition to the TPH-containing dosage forms described above, the technique has been explored for the production of a variety of drug delivery systems, such as tablets with high drug-loading [18], gastric floating core-shell structured tablets with sustained-release [19], ophthalmologic patches with controlled drug release [20], cereal-based tablets for pediatric use [21], solid lipid tablets [22], chewable tablets [23–26], personalized orodispersible films [27–29], and suppositories [30]."
Point 2: The authors did not state if they changed any parameters in the 3D printer compared to the previous study.
Response 2: On rows 185-186 (updated manuscript) we stated that "In the browser-based Brinter printing software, the print settings were set to the same as in the previous study.." and on rows 434-439 we stated that "The optimal print settings found in the previous study [26] were also suitable for the current study. As observed in the rheological results between the two studies, the TPH ink exhibited a higher viscosity, and the conclusion was drawn that a higher printing pressure would be required. As expected, a higher pressure was required, as the pressure applied for printing therapeutic doses of TPH was 1500 mbar, compared to 290 mbar for the GBP ink." To further clarify that no other parameters were changed, we have now also added "All parameters, except the printing pressure, were kept the same as in the previous study." on rows 188-189 and "...all other parameters were kept constant." on row 439.
Point 3: In terms of originality, it does not add anything new because they reuse an ink that they had already developed in a previous work, and only change the drug. In addition, they use the same printing technique, and their dosage forms have the same application as in their previous work. However, their concept of developing an ink that can be used for any drug could be interesting.
Response 3: We appreciate the reviewer’s comment, and we agree that the concept of developing an ink that can be used for several drugs is interesting. Currently, both gabapentin (previous study) and theophylline (current study) are being manually compounded in pharmacies in order to provide suitable dosage forms for veterinary use. Hence, we investigated if the same ink base could be used for both drugs in order to provide a simple but more automated compounding approach. Furthermore, this study investigated the stability of the ink as well as the stability of the printed dosage forms. The stability was not studied in the previous article, and we find this information important and necessary for the implementation of the said technique in a pharmacy or animal clinic setting. Hence, investigating both drugs is important, and the ability to use the same ink base for various drugs simplifies the overall approach, providing relevant and valuable information.
Reviewer 2 Report
The present article, by Erica Sjöholm et al, entitled: Compounding Tailored Veterinary Chewable Tablets Close to the Point-of-Care by Means of 3D Printing, refers to the authors studies on the use of semi-solid extrusion (SSE)-based 3D printing as an automated approach to the on-demand production of tailored Theophylline ChewTs (TPH), suitable for veterinary application, close to the point-of-care.
The article is nicely written, cites all the relevant literature work, the methodologies used quite appropriate and the results, which are well documented can serve as a guide for other areas, where dose tailoring is required, such as in pediatrics.
Author Response
Response to Reviewer 2 Comments
The authors appreciate the comments made by the reviewer. Per the comments provided by Reviewer 1, we have updated the bibliography including published papers from 2020 to date where SSE 3D printing was explored for the production of various drug delivery systems.
The following text has been added to the manuscript on rows 83-90 "Lately, SSE 3D printing has gained interest as a manufacturing technique for the production of drug-loaded dosage forms. In addition to the TPH-containing dosage forms described above, the technique has been explored for the production of a variety of drug delivery systems, such as tablets with high drug-loading [18], gastric floating core-shell structured tablets with sustained-release [19], ophthalmologic patches with controlled drug release [20], cereal-based tablets for pediatric use [21], solid lipid tablets [22], chewable tablets [23–26], personalized orodispersible films [27–29], and suppositories [30]."